# Corneal Adhesion Possesses the Characteristics of Solid and Membrane

**DOI:** 10.3390/bioengineering9080394

**Published:** 2022-08-16

**Authors:** Jiajin Yang, Qiaomei Ren, Dong Zhao, Zhipeng Gao, Xiaona Li, Rui He, Weiyi Chen

**Affiliations:** 1Department of Biomedical Engineering, Taiyuan University of Technology, Taiyuan 030000, China; 2State Key Laboratory of Traction Power, Southwest Jiaotong University, Chengdu 610000, China; 3Department of Excimer Laser, Shanxi Medical University, Taiyuan 030000, China

**Keywords:** biomechanics, ophthalmology, JKR theory, adhesion behavior

## Abstract

Adhesion behavior usually occurs in corneas associated with clinical treatments. Physiologically, an intact natural cornea is inflated by intraocular pressure. Due to the inflation, the physiological cornea has a mechanical property likeness to membrane. This characteristic is ignored by the classical theory used to analyze the adhesion behavior of soft solids, such as the Johnson–Kendall–Roberts (JKR) model. Performing the pull-off test, this work evidenced that the classical JKR solution was suitable for computing the corneal adhesion force corresponding to the submillimeter scale of contact. However, when the cornea was contacted at a millimeter scale, the JKR solutions were clearly smaller than the related experimental data. The reason was correlated with the membranous characteristic of the natural cornea was not considered in the JKR solid model. In this work, the modified JKR model was superimposed by the contribution from the surface tension related to the corneal inflation due to the intraocular pressure. It should be treated as a solid when the cornea is contacted at a submillimeter scale, whereas for the contact at a larger size, the characteristic of the membrane should be considered in analyzing the corneal adhesion. The modified JKR model successfully described the adhesion characteristics of the cornea from solid to membrane.

## 1. Introduction

The cornea, a transparent organ, forms the anterior pole of the eye, and it plays an essential role in visual function [1]. Adhesion behavior by the cornea is often accompanied with ophthalmic clinical treatments such as wearing contact lenses for correcting myopia [2,3]. After refractive surgeries, the stabilization of the postoperative cornea depends on the corneal cap firmly adhering to the residual stromal bed [4]. Especially, in terms of corneal transplantation, adhesion is a prominent issue of concern regarding biocompatibility; poor interfacial adhesion can affect the clinical outcome of the transplantation [5]. Studying the corneal adhesion, therefore, can help us to preferably understand the biomechanics of the cornea associated with the clinical ophthalmology.

Adhesion behavior is usually determined through a pull-off test, that is, a rigid punch indents onto a solid material and then detaches from it [6,7,8,9,10]. To analyze the adhesion behavior of a solid, the work of adhesion ***γ***, a characteristic of the material, must be considered. There are two classical theories (i.e., the Johnson–Kendall–Roberts (JKR) [11] and Derjaguin–Muller–Toporov (DMT) [12] models) used to analyze the adhesion behaviors of the solids. The DMT and JKR models describe two different types of separation. They derive two different mathematic formulas used to calculate the adhesion force, i.e., 1.5***γπR*** and 2***γπR***, with ***R*** being the derived radius. The DMT model, related to a strength-limited solution, is suitably used to study brittle failure, whereas the JKR model, related to an energy-limited solution, is suited to analyzing ductile failure [6,9,13]. Due to the fact of excellent compliance, many previous studies have evidenced that the JKR model is suitable for analyzing the adhesion behaviors of soft materials [14,15,16], including cancer cells [17] and biological soft tissues [18].

Indeed, employing the JKR model, recently Zhu and colleagues studied and compared the adhesion interactions between the cornea and silicone contact lenses of different types [3]. Their work focused on the interfacial adhesion interaction between the two different soft materials and, thus, the obtained adhesion force was coupled together with contributions from the two soft materials. It is difficult to decouple the adhesion force of the cornea itself from the interaction associated with the two soft materials. Without decoupling, this work performed the pull-off test to study corneal adhesion, employing rigid punches which had a stiffness that exceeded the cornea by many orders of magnitude.

## 2. Materials and Methods

### 2.1. Material Preparation

One hundred natural porcine eyes were consecutively collected from a local slaughterhouse, and they were immediately transported to the laboratory within two to three hours after slaughter. In the laboratory, according to the different sizes of the spherical rigid punches, the eyes were divided into seven groups (Appendix A) to study the scale effect of corneal adhesion through the pull-off test.

### 2.2. Experimental Protocol

The pull-off test, in this study, was performed at room temperature to determine the adhesion behavior of the cornea itself. Each eye specimen was indented by rigid spherical punches (elastic modulus: ≈200 GPa) with different sizes (Figure 1a and Appendix A). Prior to the indentation test, we removed the other tissues attached to the ocular surface including the extraocular muscles and fats with a pair of scissors and a tweezer at room temperature. As shown in Figure 1a, an eye specimen, with its cornea exposed to the air, was attached to a fixed chamber and fully filled with phosphate-buffered saline (PBS) by negative pressure through an injector, which could simultaneously tune the value of the IOP ***P***. We measured the adjusted intraocular pressure with a rebound tonometer (FA800vet, Shanghai, China; resolution: ±1.5–2.0 mmHg) and then measured it again after standing for 30 s. If the intraocular pressure value did not change, the intraocular pressure value of this sample could be determined [10].

Prior to the pull-off test, the IOP value was measured through a rebound tonometer with high interobserver reliability. An Instron 5544 tester (Instron, Boston, USA) with a load cell of 5 N was employed to perform the test. The corresponding resolutions of the force and the displacement were 0.001 N and 0.001 mm, respectively. During the test, a spherical rigid punch was employed to indent the corneal tissue to 2 mN and then detached from it by 1.0 mm/min (Figure 1b). The entire experimental course was carefully observed by capturing images using a DP71 (Olympus, Tokyo, Japan) charge-coupled device (CCD).

Previous studies have defined adhesion behavior. As shown in Figure 1b, it begins at the moment the punch unloads to zero force [19,20,21] and its whole course can be divided into two steps, i.e., the pull-in and the pull-off. The punch detaches from the contacted cornea to the peak value of the separation force and continues separating from the peak force to the zero force again. The peak value of the separation force is defined as the pull-off force, i.e., the adhesion force, ***F_ad_***.

### 2.3. Theoretical Analysis

Hertz (1896) first solved the contact problem between two rigid spheres [22]. On the basis of the Hertz contact pressure and indentation theory, derived by Sneddon (1965) [23], Johnson et al. (1971) [6] reported the solution of the contact between a rigid sphere and a compliance solid (Appendix A), i.e., the JKR model (derivations in the Appendix A). The related magnitude of the adhesion force solved by the JKR model was 1.5***γπR***, with 1/***R*** = 1/***R_c_*** + 1/***R_p_***; in this study, ***R_c_*** and ***R_p_*** were the radii of the cornea and the punch, respectively (Figure 1a).

Physiologically, the surrounding of a natural cornea is inflated by the IOP. The cornea, therefore, cannot be considered as an absolute solid (Figure 1c); however, it should be suitably considered to superimpose the contributions of the characteristics of a membrane (Figure 1d and Appendix A). This means when analyzing corneal adhesion, the contribution from the surface tension cannot be ignored. Accordingly, in this study, we modified the JKR model through the superposition of the terms of the surface tension [24], which was formulated as follows:(1)$$F=\frac{4E^{*}a^{3}}{3R}-\sqrt{8\pi \gamma E^{*}a^{3}}-\alpha P\pi a^{2}\frac{R_{c}}{R}$$
where ***a*** denotes the contact radius between the spherical rigid punch and the soft cornea; *P* is the intraocular pressure; and ***E**** ≈ ***E_c_***/(1 − ***υ_c_***^2^) is the effective elastic modulus, with ***E_c_*** and ***υ_c_*** being the elastic modulus and Poisson’s ratio of cornea, respectively. It was assumed that the cornea was incompressible and, thus, ***υ_c_*** was equal to 0.5. Additionally, ***α*** is a constant coefficient used to describe the degree of the cornea trending to a membrane. The detailed derivation procedure is shown in the Appendix A.

To obtain the corneal adhesion force, ***F_ad_***, in mathematics, one needs to solve the peak value from Equation (1). The definition of ***F_ad_*** is quoted from previous works [25,26], which was used to describe the maximum interfacial force (i.e., pull-off force in the Appendix A), while the punch is detached from the soft material. To determine the ***F_ad_*** value, in mathematics, one must find the critical contact radius, ***a_c_***, in the partial differential solution of ∂F/∂a=0. Then, ***a_c_***
*is* substituted into Equation (1), and the calculations result in the obtained ***F_ad_*** value.

In terms of Equation (1), while ***α***
*is* equal to zero, this describes the classical JKR solution, consisting of two terms, that is, the Hertz’s contact pressure [22] and the Kendall’s interfacial adhesion force [27]. In this case, solving the differential equation of ∂F/∂a=0, one can obtain the value of ***a_c_*** equal to ${\left(9/8\right)}^{1/3}{\left(\gamma \pi R^{2}/E^{*}\right)}^{1/3}$. Employing ***a_c_*** into Equation (1), with ***α*** = 0, one can solve the ***F_ad_*** value as 1.5***γπR***. On the other hand, while ***α*** is equal to one, Equation (1) describes the adhesion behavior of a pure membrane material. With the modified JKR model, 0 < ***α*** < 1, it is hard to obtain analytic solutions. In this study, we obtained the related numerical solutions through the software PyCharm v.3.3 (JetBrains, Prague, Czech).

To solve the magnitudes of the corneal adhesion, two intrinsic characteristics of the cornea, ***γ*** and ***E^*^***, should be determined. According to previous studies [8,9], if a compliance material exhibits linear adhesion behavior, as shown in the inset of Figure 1b, ***γ*** can be simply formulated as ***γ*** = ***σ**_m_**δ_m_***/2, where ***σ****_m_* and ***δ_m_*** are the maximum adhesion stress and distance, respectively. In terms of ***E^*^***, it can be solved through the JKR solution with zero loads, i.e., the Appendix A, which is formulated as Equation (2):(2)$$a_{0}^{3}=\frac{3R}{4E^{*}}\left(6\gamma \pi R\right)$$
where ***a*_0_** is the contact radius between the punch and the cornea at the moment the experimental data transform from the unloading phase into the pull-in phase (Figure 1b).

### 2.4. Data Analysis

The related data are presented as the mean ± standard deviation (SD), and they were statistically analyzed through one-way analysis of variance (ANOVA) subjected to an LSD test with the help of SPSS v.24.0 (SPSS Inc., Chicago, IL, USA). A probability value (*p*) less than 0.05 was considered statistically different.

## 3. Results

### 3.1. Corneal Adhesion Was Not Affected by IOP

The experimental results indicated that the corneal adhesion force, ***F_ad_***, did not vary with the IOPs. As shown in Figure 2a, there were no statistical differences in the ***F_ad_*** values (*p* > 0.05) obtained by the pull-off test during the cornea under the different IOPs in vitro, except for the comparison between the groups of 40 and 60 mmHg obtained through the 1 mm ***R_p_*** punch.

Interestingly, the related calculations of the two parameters, ***γ*** and ***E****, were similar to ***F_ad_***. As shown in Figure 2b,c, regarding punches with a certain radius, the values of ***γ*** and ***E^*^*** also did not vary with the IOPs. There were no statistical differences among the different IOP groups (*p* > 0.05). This phenomenon indicates that the parameters of ***γ*** and ***E****, obtained here, possessed the intrinsic characteristics of the cornea.

### 3.2. Scale Effect of Corneal Adhesion

To analyze the scale effect of corneal adhesion, in this study, 20 and 30 mmHg were selected as the internal environment of cornea tolerated, because the normal IOP ranged approximately from 10 to 30 mmHg [28]. Under a normal IOP of 20 or 30 mmHg, as shown in Figure 3, scale effects obviously existed in the corneal adhesion. The parameters ***γ*** and ***E^*^*** exhibited a similar tendency with the function of the punch sizes (Figure 3a,b). For example, under an IOP of 20 mmHg, regarding the punches with submillimeter sizes (i.e., ***R_p_*** ≤ 1 mm), both parameters trended towards a decrease with the increase in ***R_p_*** (*p* < 0.05). When ***R_p_*** rose to greater than 1 mm, their values trended towards a lack of statistical difference (*p* > 0.05), but they were significantly smaller than those obtained by the punch at the submillimeter scale (*p* < 0.01).

Additionally, the punch sizes could also assuredly affect the ***F_ad_***. With ***R_p_*** in the range from 0.5 to 5 mm, used here, the experimental data for ***F_ad_*** trended towards a gradual rise with the increase in ***R_p_*** (Figure 3c). However, the tendency was very slight when the punch sizes were in the range from 2 to 3 mm (*p* > 0.05). Under an IOP of 20 mmHg, for example, the ***F_ad_*** values obtained through the punches with submillimeter sizes were obviously significantly smaller than those obtained through the larger punch sizes (*p* < 0.001). The minimum value, 1.25 ± 0.25 mN, was obtained by the 0.5 mm ***R_p_*** punch. It was smaller (1.73 ± 0.39 mN) than that obtained through the 1 mm ***R_p_*** punch by approximately 28% (*p* < 0.01), and it was twofold smaller than that of the maximum value (3.94 ± 0.73 mN) obtained through the 5 mm ***R_p_*** punch (*p* < 0.001). The related data are summarized in detail in Table 1.

### 3.3. Analysis Employing the JKR Models

Substituting the obtained values of ***γ*** and ***E^*^*** into the classical and modified JKR models, we calculated the ***F_ad_*** values and compared them with the experimental data. The comparisons showed that the classical JKR model should not be used to adequately describe corneal adhesion, except for contact that has occurred at the submillimeter scale.

The experimental data for ***F_ad_***, as shown in Figure 4a, were well fitted using the classical JKR solutions when the cornea contacted with the 1 mm ***R_p_*** punch. However, for the punches with larger sizes (Figure 4b–d), such as the 3 mm ***R_p_***, the ***F_ad_*** values calculated by the JKR model were much smaller than the experimental data when the corneas had higher IOPs than normal levels (Figure 4c). Under an IOP of 40 mmHg, for example, the ***F_ad_*** value calculated using the JKR model (1.90 ± 0.76 mN) was smaller than the experimental data (2.94 ± 0.47 mN) by approximately 35% (Figure 4c). These errors were compensated for using the modified JKR solutions (Figure 4b–d).

The related scale variations described and shown in Figure 4e,f and Appendix A could evidence the tendency of corneal adhesion. Compared with the experimental data, both the classical and the modified JKR solutions obtained similar tendencies as functions of ***F_ad_*** and ***R_p_***. Unlike the modified JKR solutions, however, the related trend obtained by the JKR solutions was lower than the experimental data. Except for the submillimeter ***R_p_***, the JKR solutions were obviously smaller than the one-to-one correspondence experimental ***F_ad_*** values. On the contrary, in terms of the ***R_p_*** in the millimeter scales, the trend displayed by the modified JKR solutions was well matched with the experimental data (Appendix A and Figure 4e,f).

## 4. Discussion

This work reports that the corneal adhesion possessed a scale effect, affected by the contact area. In this study, the classical JKR model, used to describe the adhesion of solids, was valid for the analysis of corneal adhesion at submillimeter sizes (Figure 4a). However, its validity decreased with the increase in the contact area. When the punch sizes were at the millimeter scale, the corneal ***F_ad_*** obtained by the JKR solutions were obviously smaller than the related experimental data (Figure 4b–f).

This contradiction was observed between the JKR solutions and the experimental data, implying that the JKR model partially failed to analyze the corneal adhesion. To understand the reason for this, we deemed that the emphasis should be on the physiology of the natural cornea, which is fully inflated by IOP, like a membrane. To analyze corneal adhesion, therefore, the contribution of the membranous characteristic should be considered. This important factor, however, is not considered in the classical JKR model.

According to the results of the comparison, it was clear that the scale effect of corneal adhesion is associated with the variation of its characteristics from a solid to membrane. In the case that the punch size is sufficiently smaller (i.e., submillimeter scale) than the cornea, the cornea should be considered as a solid, and the classical JKR model could obtain the suitable solution (Figure 4a). Whereas when the punch size was within the millimeter scale, the cornea increasingly trended towards the characteristic of a membrane. Thus, a suitable solution for corneal adhesion should be obtained by the modified JKR model proposed here.

To simply describe the scale effect of corneal adhesion, a dimensionless normalization analysis was employed here. The factors related to corneal adhesion contained adhesion force, work of adhesion, elastic modulus, geometry, IOPs, etc. With these factors taken into account, this work proposed a parameter, ***κ***, formulated as Equation (3), to describe the scale effect.
(3)$$\kappa ={\left(\frac{F_{ad}\gamma }{PE^{*}a_{c}^{3}}\right)}^{\frac{1}{3}}$$

The parameter ***κ***, in this study, was described as a function of the ratio between the radii of the punch and the cornea: ***R_p_****/**R_c_***. The average value of the corneal radius used here was 9.58 ± 0.22 mm. Compared with the tendencies described in Figure 4e,f, interestingly, the normalization tendencies, as a function of ***κ*** and ***R_p_****/**R_c_***, exhibited the opposite (Figure 5). As shown in Figure 5, the ***κ*** values decreased with the increase in the ratio ***R_p_****/**R_c_***, and the related data were well fitted by an exponential function. For example, in terms of the cornea under the normal IOP of 20 mmHg (Figure 5a), while the punch was infinitesimal (i.e., the ratio of ***R_p_****/**R_c_*** boundlessly approaches to zero), the ***κ*** value was increasingly closer to 1.05, and the corneal adhesion increasingly trended toward the characteristic of a solid. Inversely, while the punch size was infinitely great, it was similar to an intact cornea contacted to the ground; the ***κ*** value was increasingly closer to approximately 0.29, and the corneal adhesion increasingly trended toward the characteristic of a membrane (Figure 5a). Although the normalization tendencies reversed to the dimensional, the related discrepancies between the experimental and theoretical data were the same. Compared with the experimental data, the classical JKR solutions also obtained a lower normalization tendency, and this error could also be offset by the modified JKR solutions (insets in Figure 5).

Consequently, this work evidenced that corneal adhesion possessed a scale effect due to the fact of its characteristics varying from a solid to a membrane. When a cornea was contacted by a smaller punch, within the submillimeter range, its adhesion could be described by the theory related to solids, i.e., the JKR model. While with greater contact areas, however, to analyze the corneal adhesion, the contribution of the surface tension needed considering. With an increasing contact area, the corneal adhesion increasingly trended toward a membrane. The related evidence was revealed in the results of the comparison, as shown in Figure 4, Figure 5 and Appendix A. Without wet conditions, this study related to corneal adhesion also supported the previous discovery that the scale effect existed in the wet adhesion of biological attachment systems [29].

Clinically, corneal adhesion usually appears with a greater contact area at the millimeter scale. In terms of refractive surgeries, for example, the optical operation diameters were in the range of 5–8 mm [30,31]. Wearing commercial contact lenses and corneal transplantation can cover the whole or partial surface of the cornea. Additionally, it is of great significance to study adhesives to repair corneal wounds. The current common closure methods, such as stromal hydration, suture, and wound sealant, have been reported [32]. It is clinically meaningful to study changes in corneal adhesion after using these methods. To understand the adhesion behavior of the cornea itself, this work provided a suitable theory by modifying the classical JKR model.

Compared with similar published works [3,33], this present work found that the ***F_ad_*** values of the cornea were smaller than the previous studies by one or two orders of magnitude. This is because the contact areas observed here, between the cornea and the punch, were much smaller than previous works related to corneas contacted with artificial corneas or contact lenses. The measured ***F_ad_*** value depended on the contact area (Figure 3). The ***F_ad_*** of the natural cornea was approximately 20 mN, related to the adhesion interaction between the Boston keratoprosthesis and corneal disk samples with a radius of 3 mm [33]. However, the maximum critical contact radius obtained here, ***a_c_***, related to the 5 mm ***R_p_*** punch, with a cornea IOP under 20 mmHg, was approximately 0.34 mm (Appendix A).

Additionally, rate dependence [34] and a time effect [35] obviously existed in the viscoelastic soft materials during the loading speed under the medium and high strain rates. However, under a quasi-static state, when the indentation speed was less than 500 μm/s, the adhesion characteristic of the tested soft material was stable [36]. In addition, Dai et al. indicated that the adhesion energy of soft matters due to the hydrodynamic force or the viscous force could be negligible when the speed was lower than 0.5 mm/s [25]. In this study, 1 mm/min (i.e., approximately 17 μm/s) was used as the indentation speed. Therefore, in this study, the viscous effects on the adhesion behavior determined here could also be negligible.

In summary, this study evidenced that corneal adhesion possessed a scale effect. It should be treated as a solid when the cornea is contacted at the submillimeter scale, whereas for contact of a larger size, the characteristic of a membrane should be considered when analyzing the corneal adhesion. The modified JKR model proposed here successfully described the adhesion characteristics of the cornea from a solid to membrane.

## 5. Conclusions

This study evidenced that corneal adhesion possessed a scale effect. It should be treated as a solid when the cornea is contacted at the submillimeter scale, whereas when the contact is of a larger size, the characteristic of a membrane should be considered when analyzing the corneal adhesion. The modified JKR model was superimposed by the contribution from the surface tension related to the characteristic of membrane. Where the coefficient ***α*** used to describe the degree of the cornea trending to membrane, while ***α*** equals zero, it describes the classical JKR solution. The modified JKR model proposed here successfully described the adhesion characteristics of the cornea from a solid to membrane.

## Figures and Tables

**Figure 1 bioengineering-09-00394-f001:**
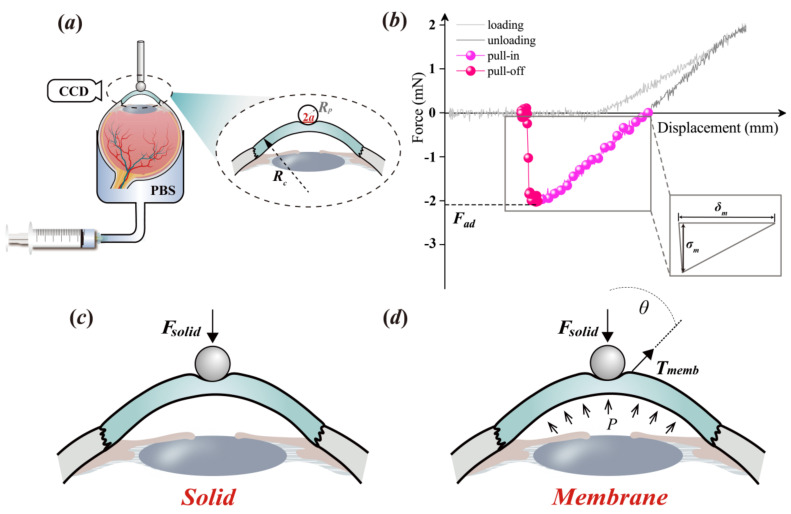
Schematic illustrations for determining corneal adhesion: (**a**) Sketch of the pull-off test in which ***R_p_*** and ***R_c_***, respectively, represent the radii of the punch and the cornea, and ***a*** is the related contact radius; (**b**) an example of the experimental data; (**c**,**d**) illustration of the related analytical theory in which ***F_solid_*** represents the adhesion force balanced by the Hertz and the Knedall terms, and ***T_memb_*** represents the surface tension related to the membrane characteristic.

**Figure 2 bioengineering-09-00394-f002:**
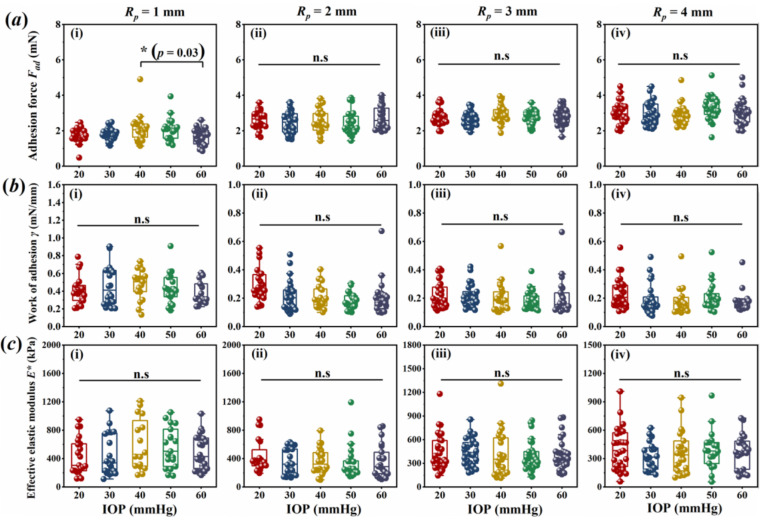
Parameters obtained from the experiment: (**a**) adhesion force (***F_ad_***); (**b**) work of adhesion ***γ***; (**c**) effective elastic modulus ***E*.*** (**n.s** indicates no statistical differences among the different groups, with *p* > 0.05 and * *p* < 0.05).

**Figure 3 bioengineering-09-00394-f003:**
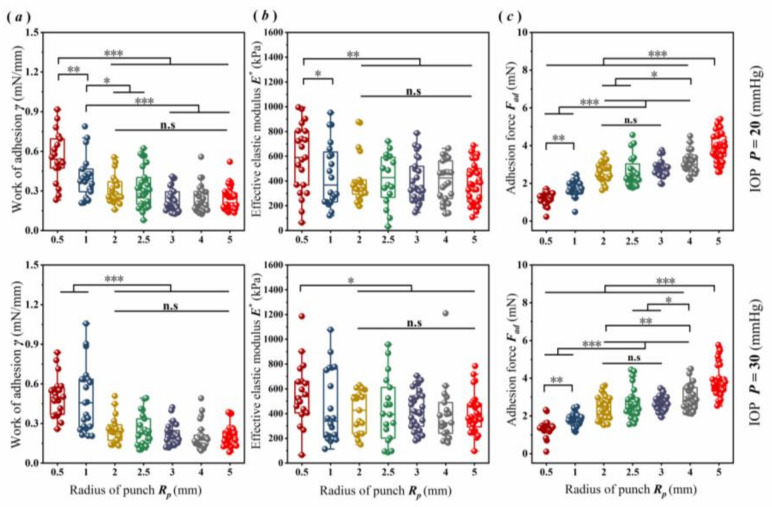
Scale effect of corneal adhesion: (**a**) work of adhesion ***γ***; (**b**) effective elastic modulus ***E^*^***; (**c**) adhesion force, ***F_ad_***, under normal IOPs of 20 and 30 mmHg, respectively (* *p* < 0.05, ** *p* < 0.01, and *** *p* < 0.001; **n.s** indicates no statistical differences among the groups, with *p* > 0.05).

**Figure 4 bioengineering-09-00394-f004:**
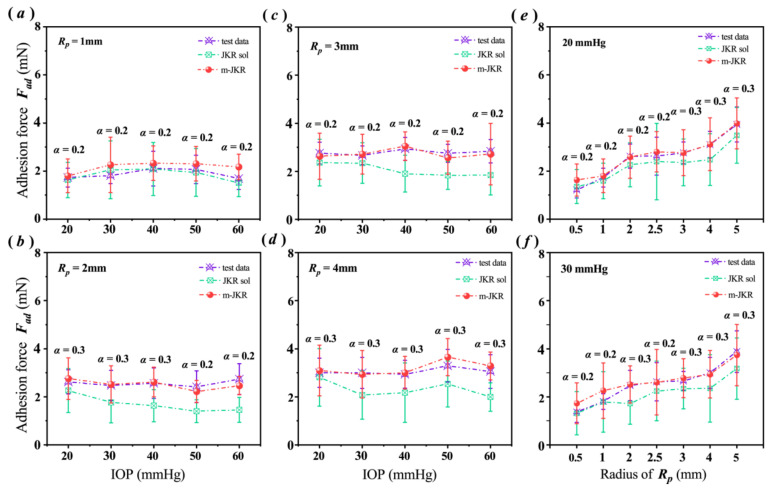
Comparisons of the corneal adhesion force, ***F_ad_***, obtained from the experiment and theories: (**a**–**d**) show comparisons of the different IOPs; (**e**,**f**) show comparisons of the scale effect of the corneal adhesion under the normal IOPs of 20 and 30 mmHg. The data are presented as the mean ± SD; JKR sol and m-JKR indicate the solutions from the JKR and modified JKR models, respectively.

**Figure 5 bioengineering-09-00394-f005:**
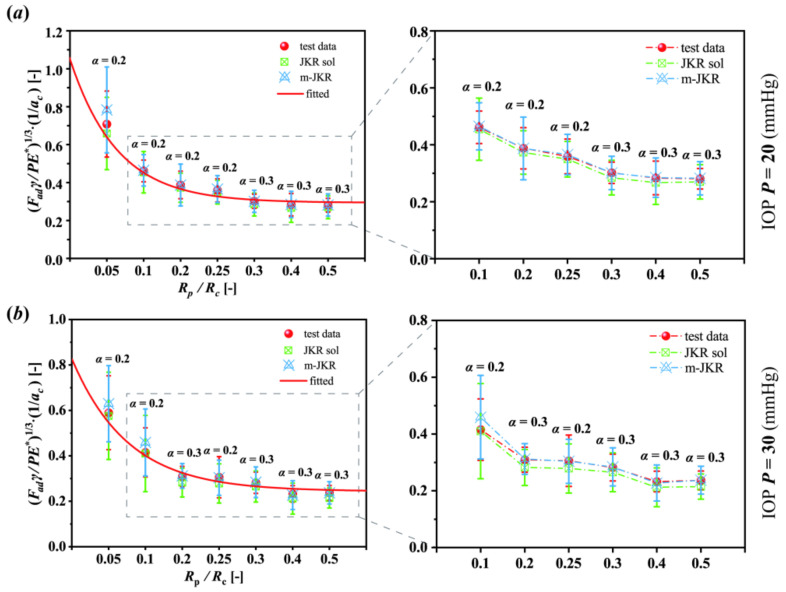
Normalization analysis of the scale effect of corneal adhesion: (**a**) under 20 mmHg, the fitting exponential function was *y* = 0.76 *exp* (−12.53*x*) + 0.29; (**b**) under 30 mmHg, the fitting exponential function was *y* = 0.58 *exp* (−10.49*x*) + 0.24. The data are presented as the mean ± SD; JKR sol and m-JKR indicate the solutions from the JKR and modified JKR models, respectively.

**Table 1 bioengineering-09-00394-t001:** The data of ***γ***, ***E****, and ***F_ad_*** obtained from the test performed here.

Radius of Punch *R_P_* (mm)								
IOP *P* (mmHg)		0.5	1	2	2.5	3	4	5
*γ* (mN/mm)	20	0.56 ± 0.19	0.41 ± 0.16	0.29 ± 0.12	0.33 ± 0.15	0.22 ± 0.09	0.23 ± 0.11	0.24 ± 0.09
	30	0.50 ± 0.15	0.46 ± 0.22	0.22 ± 0.11	0.27 ± 0.15	0.22 ± 0.08	0.19 ± 0.11	0.21 ± 0.08
	40		0.47 ± 0.17	0.21 ± 0.08		0.20 ± 0.10	0.18 ±0.09	
	50		0.43 ± 0.17	0.18 ± 0.06		0.18 ± 0.06	0.21 ± 0.10	
	60		0.37 ± 0.13	0.21 ± 0.12		0.20 ± 0.13	0.18 ± 0.08	
*E** (kPa)	20	581.52 ± 275.10	434.18 ± 260.89	437.73 ± 231.14	777.24 ± 500.72	434.92 ± 235.09	407.72 ± 224.77	381.59 ± 161.94
	30	560.80 ± 255.80	449.18 ± 292.02	361.16 ± 184.86	713.89 ± 382.38	441.29 ± 174.80	328.57 ± 145.96	403.92 ± 160.92
	40		601.36 ± 365.53	360.25 ± 182.63		422.27 ± 285.37	363.36 ± 224.58	
	50		558.01 ± 282.76	357.27 ± 247.15		393.64 ± 185.03	395.25 ± 205.55	
	60		492.78 ± 243.71	366.89 ± 237.06		418.69 ± 200.53	377.33 ± 177.10	
*F_ad_* (mN)	20	1.25 ± 0.25	1.73 ± 0.39	2.62 ± 0.50	2.62 ± 0.79	2.76 ± 0.46	3.00 ± 0.61	3.94 ± 0.73
	30	1.37 ± 0.43	1.81 ± 0.32	2.47 ± 0.62	2.62 ± 0.79	2.66 ± 0.39	3.00 ± 0.64	3.88 ± 0.87
	40		2.10 ± 0.73	2.56 ± 0.64		2.94 ± 0.47	2.94 ± 0.58	
	50		2.07 ± 0.59	2.42 ± 0.66		2.75 ± 0.38	3.29 ± 0.68	
	60		1.69 ± 0.45	2.74 ± 0.64		2.84 ± 0.49	3.05 ± 0.70	

## Data Availability

All data generated or analyzed during this study are included in this published article (and its Appendix A).

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
