# Peer review of "Corneal Adhesion Possesses the Characteristics of Solid and Membrane"

_bioengineering, 2022, doi:10.3390/bioengineering9080394_

Round 1

Reviewer 1 Report

The author proposed a modified-JKR model to ivestigate the corneal adhesion force corresponding to the different scale of contact and concluded there was a scale effect of the corneal adhesion. This is an important and well performed study that will be of great interested to the readership of your journal. However, the manuscript cannot be accepted in its current state, and there are still the following important issues that have not been resolved before.

1. Where are the references cited in the paper? which sentence? Especially, in the introduction.

2. Please give the sources of the used materials.

3. Clinically, corneal adhesion usually appears with a greater contact area at the millimeter scale. In terms of refractive surgeries, for example, the optical operation diameters were in the range of 5–8 mm. Perhaps it is more clinically meaningful to study the changes of corneal adhesion after repairing corneal wounds with adhesives. For exemple some classic wounds, 2.6-mm and 3.2-mm incisions, after common closure methods: stromal hydration, suture and wound sealant reported in some refs,  such as Chinese Chemical Letters, 2022, 33(9): 4321-4325. Please add some discussion.

4. As shown in Figure 1a, an eye specimen, with its cornea exposed to the air, was attached to a fixed chamber, fully filled with phosphate buffer saline (PBS) by negative pressure through an injector, which could simultaneously tune the value of the IOP P. Please verify the rationality of this method to provide different IOP, or cite the refs to support it. 

5. Some spelling mistakes and grammar problems, for example, "Prior to the pull-of test", 

Reviewer 2 Report

The authors of the manuscript “Corneal adhesion possesses the characteristics of solid and membrane” proposed a modified JKR model to describe the adhesion characteristics of the cornea from solid to membrane. I found the manuscript to be valuable, with appropriate evaluations and valid results. However, some changes are required.

1. The authors should elaborate on the JKR model modification in the abstract.

2. A concluding statement is required in the abstract.

3. "Pull-off" is not an appropriate keyword for this study. It should be removed or changed to "pull-off test". "Work of adhesion" should also be replaced with "adhesion behavior".

4. The authors provided references in a non-scientific manner. Please prepare the references in accordance with the "Guidelines for authors".

5. Some of the references are not up-to-date. To discuss the obtained results, the authors should refer to the most recent published papers.

6. Citations are required for all equations used in this study.

7. "et al." and "in vitro" should be written in italics throughout the manuscript.

8. The authors should elaborate on the modified JKR model just before the final sentence of the "conclusion" section.
